# Dynamic Surface Attitude Tracking Control for a Quadrotor Using Disturbance Observer

1st Tianpeng Huang
*School of Electrical Engineering and Information*
*Southwest Petroleum University*
Chengdu, China
tphuang@uestc.edu.cn

2nd Huishuang Shao*
*School of Electromechanical and Automotive Engineering*
*Yantai University*
Yantai, China
huishuangshao@163.com

*Abstract*—In this paper, a dynamic surface control strategy based on disturbance observer is proposed to stabilize three attitude angles of a quadrotor unmanned aerial vehicle (UAV) in the presence of external disturbance. To compensate for external disturbance, a disturbance observer with exponential convergence is developed. Considering the problem of explosion of complexity in traditional backstepping control, a first-order low-pass filter is used to avoid repeated differentiation of virtual control signal. Based on the disturbance observer and the filter, a robust attitude dynamic surface trajectory tracking control technique is designed for a quadrotor. The convergence of disturbance observation error and tracking error is proved using Lyapunov theory. The comparative simulations are carried out to demonstrate the effectiveness of the proposed control scheme.

*Index Terms*—Quadrotor UAV, Dynamic surface control, First-order low-pass filter, Disturbance observer, Attitude tracking

## I. INTRODUCTION

Recently, the attitude stabilization of quadrotor has attracted the attention of researchers. Many control methods have been proposed to control attitude of a quadrotor, which can be divided into two categories, i.e. model-free control and model-based control. In the model-free control scheme of quadrotor, proportional-integral-differential (PID) control [1] and active disturbance rejection control (ADRC) [2] are two typical control algorithms. In fact, ADRC is an innovation of PID, which is first proposed by Jingqing Han [3]. Compared with PID, internal and external disturbances can be estimated and compensated in ADRC. Besides, in [4], a model-free control technique based on attractive ellipsoid method is presented to achieve attitude tracking for a quadrotor. In [5], a model-free event-triggered control scheme using reinforcement learning algorithm is developed to stabilize three attitude angles of a quadrotor. In model-based control of quadrotor, some popular control techniques, e.g. linear quadratic regulator (LQR) [6], sliding mode control [7], $H_\infty$ control [8] and model predictive control [9], have been introduced to realize attitude trajectory tracking for a quadrotor. In these model-based control methods, backstepping control is a widely used control strategy in four-rotor system. In [10]–[13], a backstepping approach is designed and applied to attitude dynamics of a quadrotor. However, there exists a problem of explosion of complexity. To address this problem, a robust dynamic surface control algorithm [14]–[17] is developed for a quadrotor.

When a quadrotor system flies outdoors, it is always adversely affected by various external disturbances, such as wind. In order to ensure the attitude stabilization of quadrotor under external disturbance, the compensation for disturbance should be considered in the design of attitude controller. Designing a disturbance observer is an effective method to compensate for external disturbance. In [15], [18], a nonlinear disturbance observer is presented to estimate disturbance acting attitude dynamics of a quadrotor. In [19], an observer-based estimator for disturbance torque is proposed to improved attitude tracking accuracy of a quadrotor system. In [7], a sliding mode observer is employed to estimate disturbance in attitude subsystem of a four-rotor drone. In [16], an extended state observer is used to deal with external disturbance in quadrotor attitude channel. By designing these observers, the disturbance of attitude dynamics of quadrotor can be estimated and compensated effectively. Therefore, the stability of attitude can be improved in the presence of disturbance.

In this paper, a disturbance observer is proposed to estimate and compensate for external disturbance. Then, a robust dynamic surface control strategy based on the disturbance estimation result is designed to stabilize three attitude angles. Such a control framework can guarantee that the desired attitude trajectory is closely followed under external disturbance.

This paper is organized as follows. The mathematical model of quadrotor is introduced in Section II. The controller design of quadrotor and stability analysis are described in detail in Section III. In section IV, the simulations are performed to highlight the effectiveness of the designed controller and the related discussion is given. The conclusion of the work is drawn in Section V.

## II. DYNAMIC MODEL OF QUADROTOR UAV

The sketch of the quadrotor UAV is shown in Fig. 1, where $\{O_e : X_e - Y_e - Z_e\}$ and $\{O_b : X_b - Y_b - Z_b\}$ represent the earth and body frames respectively. Let $[\phi\ \theta\ \varphi]$ represent attitude output of the quadrotor, where $\phi$ is the roll angle, defined as the angle between the projection of the axis $O_bZ_b$ of the body frame system on the $O_e - Z_e - X_e$ plane of the earth frame system and the axis $O_eZ_e$; $\theta$ is the pitch angle, defined as the angle between the projection of the axis $O_bY_b$ of the body frame system on the $O_e - Y_e - Z_e$ plane of the

earth frame system and the axis $O_eY_e$; $\varphi$ is the yaw angle, defined as the angle between the projection of the axis $O_bX_b$ of the body frame system on the $O_e - X_e - Y_e$ plane of the earth frame system and the axis $O_eX_e$.

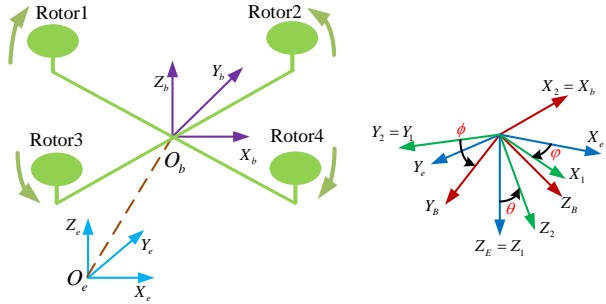

Fig. 1.  Quadrotor UAV

The attitude dynamical model of the quadrotor subject to external disturbance can be expressed by [8]

$$
\begin{aligned}
\ddot{\phi} &= \frac{(J_y - J_z)\dot{\theta}\dot{\varphi} + J_r\dot{\theta}\Omega + LF_\phi}{J_x} + d_\phi, \\
\ddot{\theta} &= \frac{(J_z - J_x)\dot{\phi}\dot{\varphi} - J_r\dot{\phi}\Omega + LF_\theta}{J_y} + d_\theta, \\
\ddot{\varphi} &= \frac{(J_x - J_y)\dot{\phi}\dot{\theta} + fF_\varphi}{J_z} + d_\varphi,
\end{aligned} \tag{1}
$$

where $J_x$, $J_y$ and $J_z$ are the moments of inertia of the quadrotor around $x$, $y$ and $z$ axes, respectively; $J_r$ is the moment of inertia of rotor; $\Omega$ is the speed difference of the diagonal rotor; $L$ is the distance from the center of the rotor to the center of the quadrotor system; $f$ is the scaling factor from force to moment. $F_\phi$, $F_\theta$ and $F_\varphi$ are the control inputs; $d_\phi$, $d_\theta$ and $d_\varphi$ are the external inputs of the quadrotor system in the form of disturbances.

It is worth mentioning that compared with brushless motor, the rotor is light. Therefore, the terms $\frac{J_r\dot{\theta}\Omega}{J_x}$ and $\frac{J_r\dot{\phi}\Omega}{J_y}$ can be ignored.

## III. CONTROL DESIGN

The following assumptions are given to make the subsequent analysis rigorous.

*Assumption 1:* It is assumed that the disturbances $d_\phi$, $d_\theta$ and $d_\varphi$ acting on the attitude channel of the quadrotor system change slowly, i.e., $\dot{d}_\phi = \dot{d}_\theta = \dot{d}_\varphi = 0$.

*Assumption 2:* Suppose that both the first derivative and the second derivative of attitude desired trajectories $\phi_d$, $\theta_d$ and $\varphi_d$ are bounded.

*Assumption 3:* It is assumed that the first derivative of attitude output $\phi$, $\theta$ and $\varphi$ is bounded.

It should be noted that, the desired trajectory is given manually. Therefore, the boundedness of its first derivative and second derivative can be guaranteed. The quadrotor UAV can not be required to track a trajectory with infinitely fast changes in speed and acceleration for any specific task. Moreover, In the actual quadrotor system, the output of the attitude

angle cannot be changed infinitely fast, which makes its first derivative bounded.

*Lemma 1* [20]: Suppose that there exists a continuous, positive definite function $V(t)$ such that

$$
\dot{V}(t) \le -\alpha V(t) + f(t), \forall t \ge t_0 \ge 0, \tag{2}
$$

where $\alpha$ is any finite constant and $f(t)$ is a function of time $t$. Then, the following inequality holds

$$
V(t) \le \underline{e}^{-\alpha(t-t_0)}V(t_0) + \int_{t_0}^{t} \underline{e}^{-\alpha(t-\tau)}f(\tau)d_\tau, \tag{3}
$$

where $\underline{e}$ is the base of the natural logarithm and $V(t_0)$ is initial value of $V(t)$.

### A. Disturbance observer design

To compensate for external disturbance, a disturbance observer with exponential convergence is introduced. Define the estimation error of disturbance observer as

$$
\bar{e}_\chi = d_\chi - \hat{d}_\chi, \tag{4}
$$

where $\chi = \phi, \theta, \varphi$ and $\hat{d}_\chi$ is the estimation of $d_\chi$. Then, the disturbance observer is designed as

$$
\begin{aligned}
\dot{\zeta}_\chi &= -\lambda_\chi\zeta_\chi - \lambda_\chi^2\dot{\chi} - \lambda_\chi P_\chi - \lambda_\chi Q_\chi F_\chi, \\
\hat{d}_\chi &= \zeta_\chi + \lambda_\chi\dot{\chi},
\end{aligned} \tag{5}
$$

where $\lambda_\chi$ is a positive constant and $\zeta_\chi$ is state variable of the observer. Moreover, $P_\chi = \frac{J_y - J_z}{J_x}\dot{\theta}\dot{\varphi}$ and $Q_\chi = \frac{L}{J_x}$ when $\chi = \phi$; $P_\chi = \frac{J_z - J_x}{J_y}\dot{\phi}\dot{\varphi}$ and $Q_\chi = \frac{L}{J_y}$ when $\chi = \theta$; $P_\chi = \frac{J_x - J_y}{J_z}\dot{\phi}\dot{\theta}$ and $Q_\chi = \frac{f}{J_z}$ when $\chi = \varphi$.

*Theorem 1:* Considering the dynamics (1) of the quadrotor, if the *Assumption 1* holds and the disturbance observer is designed as (5), then the disturbance estimation error (4) will converge to zero as $t \to \infty$ exponentially.

*Proof 1:* Differentiating $\hat{d}_\chi$ in (5), it is obtained

$$
\dot{\hat{d}}_\chi = \dot{\zeta}_\chi + \lambda_\chi\ddot{\chi}. \tag{6}
$$

Substituting the first equation in (5) into (6), one has

$$
\dot{\hat{d}}_\chi = -\lambda_\chi - \lambda_\chi^2\dot{\chi} - \lambda_\chi P_\chi - \lambda_\chi Q_\chi F_\chi + \lambda_\chi\ddot{\chi}. \tag{7}
$$

Applying (1) to (7) yields

$$
\begin{aligned}
\dot{\hat{d}}_\chi &= -\lambda_\chi\zeta_\chi - \lambda_\chi^2\dot{\chi} + \lambda_\chi d_\chi \\
&= -\lambda_\chi(\hat{d}_\chi - d_\chi) \\
&= \lambda_\chi\bar{e}_\chi.
\end{aligned} \tag{8}
$$

Meanwhile, according to *Assumption 1* and (4), the dynamics of disturbance observation error can be expressed by

$$
\begin{aligned}
\dot{\bar{e}}_\chi &= \dot{d}_\chi - \dot{\hat{d}}_\chi \\
&= -\dot{\hat{d}}_\chi.
\end{aligned} \tag{9}
$$

Combining (8) and (9), we have

$$
\dot{\bar{e}}_\chi = -\lambda_\chi\bar{e}_\chi. \tag{10}
$$

Solving (11), one has

$$\bar{e}_\chi = e^{-\lambda_\chi t}\bar{e}_\chi(0). \tag{11}$$

where $\bar{e}_\chi(0)$ is initial value of observation error. Obviously, the disturbance observation error in attitude channel has a property of exponential convergence.

This completes the proof. ∎

### B. Attitude controller design

Define tracking error as

$$e_{\chi 1} = \chi_d - \chi, \tag{12}$$

where $\chi_d$ is desired attitude signal. To avoid repeated differentiation of virtual control signal, define the following first-order low-pass filter in Laplace domain

$$\xi_\chi = \frac{1}{T_\chi s + 1}\gamma_\chi, \tag{13}$$

where $s$ is a variable in Laplace domain; $T_\chi$ is a filter time constant; $\xi_\chi$ is output of the filter; $\gamma_\chi$ is input of the filter, defined as

$$\gamma_\chi = -c_{\chi 1}e_{\chi 1} - \dot{\chi}_d, \tag{14}$$

where $c_{\chi 1}$ is a positive constant. The expression of (13) in time domain can be further written as

$$\dot{\xi}_\chi = \frac{\gamma_\chi - \xi_\chi}{T_\chi}. \tag{15}$$

The attitude controller is designed as

$$F_\chi = \frac{1}{Q_\chi}(-P_\chi - \dot{\xi}_\chi + c_{\chi 2}e_{\chi 2} - \hat{d}_\chi), \tag{16}$$

where $c_{\chi 2}$ is a positive constant and $e_{\chi 2}$ is defined as

$$e_{\chi 2} = -\xi_\chi - \dot{\chi}. \tag{17}$$

*Theorem 2:* For the attitude dynamics of the quadrotor in (1), if the disturbance observer is designed as (5) and the controller is chosen as (16), then the attitude tracking error (12) of the closed-loop system will be guaranteed to converge to zero asymptotically.

*Proof 2:* Define the following Lyapunov function candidate

$$V_\chi = \frac{1}{2}e_{\chi 1}^2 + \frac{1}{2}e_{\chi 2}^2 + \frac{1}{2}e_{\xi\gamma\chi}^2, \tag{18}$$

where $e_{\xi\gamma\chi}$ is filter error, defined as

$$e_{\xi\gamma\chi} = \xi_\chi - \gamma_\chi. \tag{19}$$

The derivative of $V_\chi$ with respect to time is

$$\dot{V}_\chi = e_{\chi 1}\dot{e}_{\chi 1} + e_{\chi 2}\dot{e}_{\chi 2} + e_{\xi\gamma\chi}\dot{e}_{\xi\gamma\chi}. \tag{20}$$

In the following, we calculate $\dot{e}_{\chi 1}$, $\dot{e}_{\chi 2}$ and $\dot{e}_{\xi\gamma\chi}$, respectively. $\dot{e}_{\chi 1}$ is first addressed. Using (12), one has

$$\dot{e}_{\chi 1} = \dot{\chi}_d - \dot{\chi}. \tag{21}$$

Substituting (17) into (21) yields

$$\dot{e}_{\chi 1} = \dot{\chi}_d + e_{\chi 2} + \xi_\chi. \tag{22}$$

Substituting (19) into (22), we have

$$\dot{e}_{\chi 1} = \dot{\chi}_d + e_{\chi 2} + e_{\xi\gamma\chi} + \gamma_\chi. \tag{23}$$

Applying (14) into (23), it is obtained that

$$\dot{e}_{\chi 1} = -c_{\chi 1}e_{\chi 1} + e_{\chi 2} + e_{\xi\gamma\chi}. \tag{24}$$

Then, $\dot{e}_{\chi 2}$ is addressed. According to (17), one has

$$\dot{e}_{\chi 2} = -\dot{\xi}_\chi - \ddot{\chi}. \tag{25}$$

Substituting (1) into (25), the expression of $\dot{e}_{\chi 2}$ can be given by

$$\dot{e}_{\chi 2} = -\dot{\xi}_\chi - P_\chi - Q_\chi F_\chi - d_\chi. \tag{26}$$

Furthermore, $\dot{e}_{\xi\gamma\chi}$ is addressed. Considering the definition of filter error (19), we get

$$\dot{e}_{\xi\gamma\chi} = \dot{\xi}_\chi - \dot{\gamma}_\chi. \tag{27}$$

Substituting (15) and (14) into (27) yields

$$\dot{e}_{\xi\gamma\chi} = \frac{\gamma_\chi - \xi_\chi}{T_\chi} + c_{\chi 1}\dot{e}_{\chi 1} + \ddot{\chi}_d. \tag{28}$$

(28) can be rewritten as from (19)

$$\dot{e}_{\xi\chi} = \frac{-e_{\xi\chi}}{T_\chi} + c_{\chi 1}\dot{e}_{\chi 1} + \ddot{\chi}_d. \tag{29}$$

Substituting (24), (26) and (29) into (20), one has

$$\begin{aligned}
\dot{V}_\chi =& e_{\chi 1}(-c_{\chi 1}e_{\chi 1} + e_{\chi 2} + e_{\xi\gamma\chi}) \\
&+ e_{\chi 2}(-\dot{\xi}_\chi - P_\chi - Q_\chi F_\chi - d_\chi) \\
&+ e_{\xi\gamma\chi}(\frac{-e_{\xi\gamma\chi}}{T_\chi} + c_{\chi 1}\dot{e}_{\chi 1} + \ddot{\chi}_d).
\end{aligned} \tag{30}$$

Applying the designed attitude controller (16) to (30), it follows that

$$\begin{aligned}
\dot{V}_\chi =& e_{\chi 1}(-c_{\chi 1}e_{\chi 1} + e_{\chi 2} + e_{\xi\gamma\chi}) \\
&+ e_{\chi 2}(-c_{\chi 2}e_{\chi 2} + \hat{d}_\chi - d_\chi) \\
&+ e_{\xi\gamma\chi}(\frac{-e_{\xi\gamma\chi}}{T_\chi} + c_{\chi 1}\dot{e}_{\chi 1} + \ddot{\chi}_d) \\
=& e_{\chi 1}e_{\chi 2} + e_{\chi 1}e_{\xi\gamma\chi} - c_{\chi 1}e_{\chi 1}^2 - c_{\chi 2}e_{\chi 2}^2 \\
&+ e_{\chi 2}(\hat{d}_\chi - d_\chi) - \frac{e_{\xi\gamma\chi}^2}{T_\chi} + e_{\xi\gamma\chi}f_\chi,
\end{aligned} \tag{31}$$

where $f_\chi = c_{\chi 1}\dot{e}_{\chi 1} + \ddot{\chi}_d$. According to *Assumption 2* and *Assumption 3*, $f_\chi$ is a bounded function. Therefore, $f_\chi$ has a maximum value, and suppose this maximum value is $M_{f_\chi}$. Moreover, the term $\hat{d}_\chi - d_\chi$ will converge to zero as $t \to \infty$ from *Theorem 1*. Obviously, (31) can be rewritten as

$$\begin{aligned}
\dot{V}_\chi =& e_{\chi 1}e_{\chi 2} + e_{\chi 1}e_{\xi\gamma\chi} - c_{\chi 1}e_{\chi 1}^2 - c_{\chi 2}e_{\chi 2}^2 \\
&- \frac{e_{\xi\gamma\chi}^2}{T_\chi} + e_{\xi\gamma\chi}f_\chi.
\end{aligned} \tag{32}$$

Applying Young's inequality to (32), it is obtained that

$$
\begin{aligned}
\dot{V}_\chi \leq & \frac{1}{2}(e_{\chi1}^2 + e_{\chi2}^2) + \frac{1}{2}(e_{\chi1}^2 + e_{\xi\gamma\chi}^2) \\
& - c_{\chi1}e_{\chi1}^2 - c_{\chi2}e_{\chi2}^2 - \frac{e_{\xi\gamma\chi}^2}{T_\chi} + e_{\xi\gamma\chi}f_\chi \\
\leq & (1 - c_{\chi1})e_{\chi1}^2 + (\frac{1}{2} - c_{\chi2})e_{\chi2}^2 + \frac{1}{2}e_{\xi\gamma\chi}^2 \\
& - \frac{e_{\xi\gamma\chi}^2}{T_\chi} + e_{\xi\gamma\chi}f_\chi.
\end{aligned}
\tag{33}
$$

Meanwhile, note that since the following inequality

$$
(e_{\xi\gamma\chi}f_\chi - c_{f_\chi})^2 \geq 0
\tag{34}
$$

always holds with $c_{f_\chi}$ being a positive constant, we have

$$
e_{\xi\gamma\chi}f_\chi \leq \frac{1}{2c_{f_\chi}}e_{\xi\gamma\chi}^2 f_\chi^2 + \frac{c_{f_\chi}}{2}.
\tag{35}
$$

Combining (33) and (35), one has

$$
\begin{aligned}
\dot{V}_\chi \leq & (1 - c_{\chi1})e_{\chi1}^2 + (\frac{1}{2} - c_{\chi2})e_{\chi2}^2 + \frac{1}{2}e_{\xi\gamma\chi}^2 \\
& - \frac{e_{\xi\gamma\chi}^2}{T_\chi} + \frac{1}{2c_{f_\chi}}e_{\xi\gamma\chi}^2 f_\chi^2 + \frac{c_{f_\chi}}{2} \\
\leq & (1 - c_{\chi1})e_{\chi1}^2 + (\frac{1}{2} - c_{\chi2})e_{\chi2}^2 \\
& + (\frac{1}{2c_{f_\chi}}f_\chi^2 + \frac{1}{2} - \frac{1}{T_\chi})e_{\xi\gamma\chi}^2 + \frac{c_{f_\chi}}{2}.
\end{aligned}
\tag{36}
$$

Let

$$
\begin{aligned}
\delta_\chi &\geq 0, \\
c_{\chi1} &\geq 1 + \delta_\chi, \\
c_{\chi1} &\geq \frac{1}{2} + \delta_\chi, \\
\frac{1}{T_\chi} &\geq \frac{1}{2c_{f_\chi}}M_{f_\chi}^2 + \frac{1}{2} + \delta_\chi,
\end{aligned}
\tag{37}
$$

then (36) can be rewritten as

$$
\begin{aligned}
\dot{V}_\chi \leq & -\delta_\chi e_{\chi1}^2 - \delta_\chi e_{\chi2}^2 \\
& + (\frac{1}{2c_{f_\chi}}f_\chi^2 - \frac{1}{2c_{f_\chi}}M_{f_\chi}^2 - \delta_\chi)e_{\xi\gamma\chi}^2 + \frac{c_{f_\chi}}{2} \\
\leq & -\delta_\chi e_{\chi1}^2 - \delta_\chi e_{\chi2}^2 - -\delta_\chi e_{\xi\gamma\chi}^2 \\
& + (\frac{1}{2c_{f_\chi}}f_\chi^2 - \frac{1}{2c_{f_\chi}}M_{f_\chi}^2)e_{\xi\gamma\chi}^2 + \frac{c_{f_\chi}}{2} \\
\leq & -2\delta_\chi V_\chi + (\frac{1}{2c_{f_\chi}}f_\chi^2 - \frac{1}{2c_{f_\chi}}M_{f_\chi}^2)e_{\xi\gamma\chi}^2 + \frac{c_{f_\chi}}{2}.
\end{aligned}
\tag{38}
$$

Let $\Gamma = (\frac{1}{2c_{f_\chi}}f_\chi^2 - \frac{1}{2c_{f_\chi}}M_{f_\chi}^2)e_{\xi\gamma\chi}^2$, we have

$$
\begin{aligned}
\Gamma = & (\frac{1}{2c_{f_\chi}}\frac{M_{f_\chi}^2 f_\chi^2}{M_{f_\chi}^2} - \frac{1}{2c_{f_\chi}}M_{f_\chi}^2)e_{\xi\gamma\chi}^2 \\
= & (\frac{f_\chi^2}{M_{f_\chi}^2} - 1)\frac{1}{2c_{f_\chi}}M_{f_\chi}^2 e_{\xi\gamma\chi}^2.
\end{aligned}
\tag{39}
$$

Since $M_{f_\chi}$ is maximum value of $f_\chi$, $\frac{f_\chi^2}{M_{f_\chi}^2} \leq 1$. Obviously, $\Gamma \leq 0$ from (39). Using (38), we have

$$
\dot{V}_\chi \leq -2\delta_\chi V_\chi + \frac{c_{f_\chi}}{2}.
\tag{40}
$$

From *Lemma 1*, the solution of (40) can be given by

$$
\begin{aligned}
V_\chi \leq & \underline{e}^{-2\delta_\chi(t-t_0)}V_\chi(t_0) + \int_{t_0}^t \underline{e}^{-2\delta_\chi(t-\tau)}\frac{c_{f_\chi}}{2}d_\tau \\
\leq & \underline{e}^{-2\delta_\chi(t-t_0)}V_\chi(t_0) + \frac{c_{f_\chi}}{2}\underline{e}^{-2\delta_\chi t}\frac{1}{2\delta_\chi}(\underline{e}^{2\delta_\chi t} - \underline{e}^{2\delta_\chi t_0}) \\
\leq & \underline{e}^{-2\delta_\chi(t-t_0)}V_\chi(t_0) + \frac{c_{f_\chi}}{2}\frac{1}{2\delta_\chi}(1 - \underline{e}^{-2\delta_\chi(t-t_0)}),
\end{aligned}
\tag{41}
$$

where $V_\chi(t_0)$ is initial value of $V_\chi$. When $t_0 = 0$, (41) can be further rewritten as

$$
V_\chi \leq \underline{e}^{-2\delta_\chi(t)}V_\chi(0) + \frac{c_{f_\chi}}{2}\frac{1}{2\delta_\chi}(1 - \underline{e}^{-2\delta_\chi t}).
\tag{42}
$$

Thereby, we get

$$
\lim_{t\to\infty} V_\chi \leq \frac{c_{f_\chi}}{4\delta_\chi}.
\tag{43}
$$

To guarantee that the attitude tracking error converges to a neighborhood of zero, the value of $\delta_\chi$ should be selected as greater as possible than that of $c_{f_\chi}$. Thus, the convergence accuracy of attitude tracking error can be improved.

This completes the proof. ∎

## IV. SIMULATION RESULTS AND DISCUSSION

In simulation, the proposed control scheme is compared with DSC without disturbance observer (DSC-WDO) and LQR. Note that in order to ensure the fairness of the tracking results, the proposed controller and DSC-WDO choose the same control gains. The parameters of the quadrotor UAV are set as follows [21]: $L = 0.4\ m$, $J_x = 0.16\ kgm^2$, $J_y = 0.16\ kgm^2$, $J_z = 0.32\ kgm^2$, $f = 0.05\ m$. All the initial attitude states of the quadrotor system are set to $0\ rad$. We consider the following two cases for desired attitude and external disturbance in simulations.

*Case 1*: The desired attitude signals are constant, and the external disturbances are time-varying, i.e. $\phi_d = \theta_d = \varphi_d = 0.32$, $d_\phi = 1.2\cos(\frac{\pi}{3}t + \frac{\pi}{6}) + 0.14$, $d_\theta = 1$ for $t \in [0,4] \cup (8,12] \cup (16,20)$, $d_\theta = 0$ for $t \in (4,8] \cup (12,16]$, $d_\varphi = 0.45t + 0.2$ for $t \in [0,5]$, $d_\varphi = 2.45$ for $t \in (5,10]$, and $d_\varphi = -0.16t + 5.45$ for $t \in (10,20]$.

The observations of the external disturbances for the three attitude channels is shown in Fig. 2, where we find there will be a small observation error using the proposed observer when the external disturbances are time-varying. However, the external disturbances can be accurately observed when they are constant.

The control inputs of the three attitude channels are depicted in Fig. 3. The corresponding tracking results of the three attitude angles are plotted in Fig. 4. In roll channel, it can be seen that when the quadrotor UAV encounters a time-varying disturbance with periodic oscillation, the tracking result produces a more obvious oscillation using LQR control scheme, while a more stable tracking result is provided by the proposed controller. In pitch channel, we observe that when the disturbance does not exist, the three control algorithms give roughly equal tracking performance, but when the disturbance

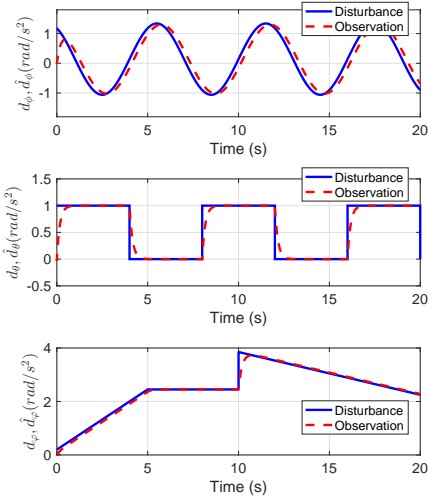

Fig. 2. The observation results of external disturbances in *case 1*.

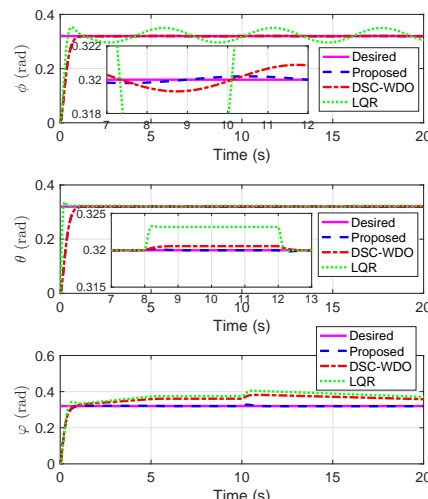

Fig. 4. The tracking results for the desired attitude in *case 1*.

occurs, such as the interval of 8s to 12s, the tracking ability of the proposed control algorithm is significantly better than that of the other two algorithms. In yaw channel, only the proposed control approach can enable the desired signal to be closely followed when the disturbance in the form of a piecewise function is injected into the quadrotor systems.

estimate the external constant disturbances. Therefore, the compensation of these disturbances can be achieved in the attitude controllers.

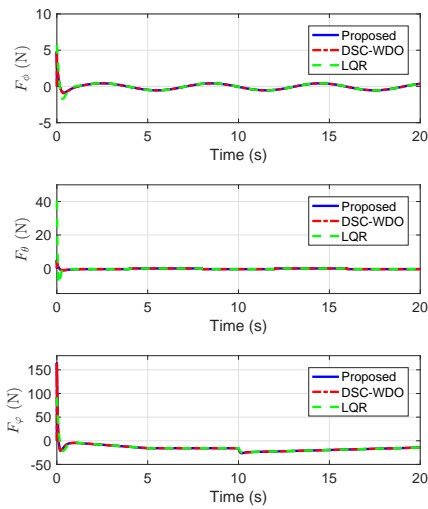

Fig. 3. The control inputs of the quadrotor in *case 1*.

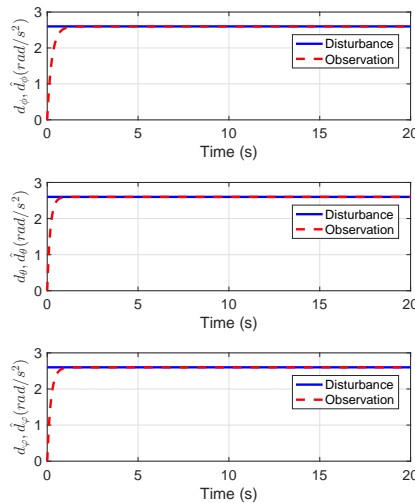

Fig. 5. The observation results of external disturbances in *case 2*.

*Case 2*: The desired attitude signals are time-varying, and the external disturbances are constant, i.e. $\phi_d = 0.015t + 0.12$, $\theta_d = 0.24\sin(\frac{\pi}{4}t)$, $\varphi_d = 0.17\cos(\frac{\pi}{3}t + \frac{\pi}{6})$ and $d_\phi = d_\theta = d_\varphi = 2.6$.

The observations of the external disturbances for the three attitude channels are given in Fig. 5. As shown in Fig. 5, the designed nonlinear disturbance observer can accurately

The control inputs of the three attitude channels are shown in Fig. 6, from which we find these control signals are similar in the steady-state stage. The tracking results for the three desired attitude angles are shown in Fig. 7. In roll channel, there is an obvious steady-state tracking error when LQR is applied to the quadrotor. Such tracking result is unacceptable in the actual application of quadrotor UAV. In pitch channel, all three control algorithms can help quadrotor drone to accurately track standard sinusoidal signal. However, when the desired standard sinusoidal signal is shifted, it can be seen that

LQR and DSC-WDO produce a larger tracking error, while the actual output can still follow the reference input using the proposed control technique.

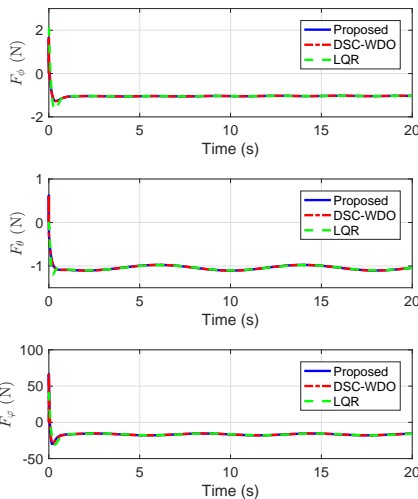

Fig. 6. The control inputs of the quadrotor in *case 2*.

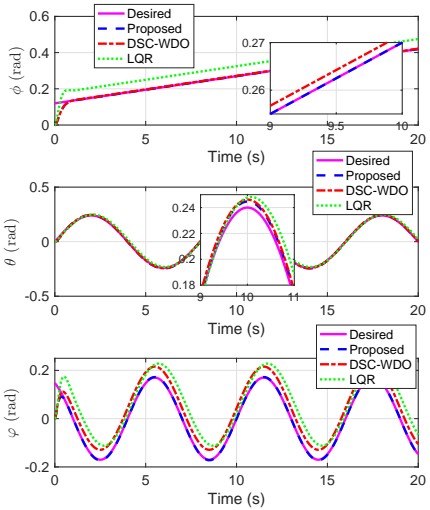

Fig. 7. The tracking results for the desired attitude in *case 2*.

## V. CONCLUSION

In this paper, a robust dynamic surface control scheme is proposed to achieve attitude tracking of a quadrotor. To compensate for external disturbance, a nonlinear disturbance observer with exponential convergence is developed to estimate external disturbance. A first-order low-pass filter is employed to address the problem of explosion of complexity in traditional backstepping. The stability of closed-loop system is rigorously proved based on Lyapunov theory. The comparative simulations are carried out to illustrate the effectiveness of the proposed control strategy.

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
