# OpenReview forum: "Dynamic Surface Attitude Tracking Control for a Quadrotor Using Disturbance Observer"
_IEEE.org/ICIST/2024/Conference — IEEE ICIST 2024 Conference Submission_

### Official Review · Reviewer_R1FG · 2024-08-22
**This article is quite fascinating and of high quality.**

**Rating:** 7
**Confidence:** 3

**Review:**

In this paper, "Dynamic Surface Attitude Tracking Control for a Quadrotor Using Disturbance Observer", a dynamic surface control method of quadrotor UAV based on disturbance observer is proposed. Firstly, in order to compensate the external interference, an exponentially convergent interference observer is proposed. Finally, aiming at the complexity explosion problem in traditional backstepping control, a first-order low-pass filter is used to avoid repeated differentiation of virtual control signals. The article has clear logic and organization, but there are still some problems. My specific feedback is as follows :1) In the introduction, the author's analysis of the current situation of the research content is insufficient. 2) In the study, what are the characteristics of the interference observer with exponential convergence in control compared with the general interference observer? 3) Please check the grammar and spelling of the words in the passage.

---

### Official Review · Reviewer_9X5d · 2024-08-22
**This article is well written and can be accepted.**

**Rating:** 7
**Confidence:** 3

**Review:**

1.To make it easier for readers to understand the novelties of this paper, it would be better for the authors to add the control frame diagram. 2.The robustness of the proposed control method should be described.
3.There are some grammatical and typographical errors in the manuscript. Please check the full text carefully and correct them.

---

### Official Review · Reviewer_FcMu · 2024-08-23
**accept**

**Rating:** 7
**Confidence:** 3

**Review:**

The paper presents a dynamic surface control strategy based on disturbance observer for stabilizing the three attitude angles of a  aerial vehicle  under external disturbances. The theory is correct and can be accepted after responding the following comments.
(1) What is the contribution of the paper? It should be highlighted both in the introduction and in the content.
(2) Please check if you need to update your Introduction.
(3)The conclusion of the article suggests using the present perfect tense for description.

---

### Decision · Program_Chairs · 2024-09-06

Accept (Oral)